# Inflammasome Activity in the Skeletal Muscle and Heart of Rodent Models for Duchenne Muscular Dystrophy

**DOI:** 10.3390/ijms24108497

**Published:** 2023-05-09

**Authors:** Zsófia Onódi, Petra Lujza Szabó, Dániel Kucsera, Péter Pokreisz, Christopher Dostal, Karlheinz Hilber, Gavin Y. Oudit, Bruno K. Podesser, Péter Ferdinandy, Zoltán V. Varga, Attila Kiss

**Affiliations:** 1Department of Pharmacology and Pharmacotherapy, Semmelweis University, 1085 Budapest, Hungary; onodi.zsofia@med.semmelweis-univ.hu (Z.O.);; 2HCEMM-SE Cardiometabolic Immunology Research Group, Semmelweis University, 1085 Budapest, Hungary; 3MTA-SE Momentum Cardio-Oncology and Cardioimmunology Research Group, Semmelweis University, 1085 Budapest, Hungary; 4Ludwig Boltzmann Institute for Cardiovascular Research at the Center for Biomedical Research and Translational Surgery, Medical University of Vienna, 1090 Vienna, Austria; 5Department of Neurophysiology & Neuropharmacology, Center for Physiology & Pharmacology, Medical University of Vienna, 1090 Vienna, Austria; 6Division of Cardiology, Department of Medicine, University of Alberta, Edmonton, AB T6G 2R3, Canada; 7Pharmahungary Group, 6728 Szeged, Hungary

**Keywords:** inflammation, heart failure, cardiomyopathy, skeletal muscle, muscular dystrophy

## Abstract

Duchenne muscular dystrophy (DMD) is characterized by wasting of muscles that leads to difficulty moving and premature death, mainly from heart failure. Glucocorticoids are applied in the management of the disease, supporting the hypothesis that inflammation may be driver as well as target. However, the inflammatory mechanisms during progression of cardiac and skeletal muscle dysfunction are still not well characterized. Our objective was to characterize the inflammasomes in myocardial and skeletal muscle in rodent models of DMD. Gastrocnemius and heart samples were collected from *mdx* mice and DMD^mdx^ rats (3 and 9–10 months). Inflammasome sensors and effectors were assessed by immunoblotting. Histology was used to assess leukocyte infiltration and fibrosis. In gastrocnemius, a tendency towards elevation of gasdermin D irrespective of the age of the animal was observed. The adaptor protein was elevated in the *mdx* mouse skeletal muscle and heart. Increased cleavage of the cytokines was observed in the skeletal muscle of the DMD^mdx^ rats. Sensor or cytokine expression was not changed in the tissue samples of the *mdx* mice. In conclusion, inflammatory responses are distinct between the skeletal muscle and heart in relevant models of DMD. Inflammation tends to decrease over time, supporting the clinical observations that the efficacy of anti-inflammatory therapies might be more prominent in the early stage.

## 1. Introduction

Duchenne muscular dystrophy (DMD) is an X chromosome-linked disease caused by various mutations in the gene encoding the dystrophin protein. The illness is generally characterized by degeneration of muscle tissue, in particular the skeletal muscle, and DMD patients also are often diagnosed with progressive dilated cardiomyopathy [1]. Self-maintaining inflammatory reactions exacerbate muscle degeneration: cellular necrosis induced by the absence of the dystrophin protein with the subsequent inflammation and excessive fibrosis contributes to a vicious cycle in the muscle tissue [2]. Thus, chronic anti-inflammatory therapy, in particular with glucocorticoids, is a number one therapeutic regimen for DMD patients [1,3].

Inflammatory signaling pathways, such as nuclear factor kappa B (NF-κB) or its downstream mediators, such as tumor necrosis factor alpha (TNFα), interleukin-6 (IL-6) or interleukin-1 beta (IL-1β), are dysregulated in DMD models [4,5]. The main regulators of inflammation are believed to be important therapeutic targets, as attenuating chronic inflammation might reduce muscle damage and fibrosis. Although disease progression is primarily caused by the deficient dystrophin protein, combined therapy for improving the phenotype of muscle cells and preventing chronic inflammation with subsequent fibrosis may be an applicable therapeutic approach for the management of DMD [6]. Several drugs, including NF-κB inhibitor edasalonexent or a steroid analogue vamorolone, have been tested in clinical trials with variable outcomes [7,8]. Therefore, identifying new pharmacological targets and anti-inflammatory strategies is essential for improving the clinical outcome for DMD.

IL-1β is a key pro-inflammatory cytokine, contributing to the progression of various diseases with inflammatory mechanisms including muscular dystrophy [9,10]. The protein expression of pro-IL-1β and other inflammasome components can be induced by several mechanisms; a process that is referred to as inflammasome priming via the NF-κB-dependent pathway [11]. Inflammasome activation is initiated by pathogen- or danger-associated molecular patterns that trigger the assembly of inflammasomes and lead to the cleavage of the caspase-1 enzyme, which activates pro-IL-1β [12]. A previous report suggests that the NLRP3 inflammasome might play a role in the pathomechanism of muscular dystrophies in a specific DMD murine model [10]. However, inflammasome priming and activation have not been investigated in DMD-associated cardiomyopathy.

In this study, we investigated inflammasome priming and activation in the heart and the skeletal muscle in two rodent (*mdx* mouse and DMD^mdx^ rat) models of DMD. To analyze the time dependent inflammatory processes, we performed the phenotyping both at early and later stages in the disease progression.

## 2. Results

### 2.1. Expression of Inflammasome Adaptor Protein Apoptosis-Associated Speck-like Protein Containing a CARD and Gasdermin D Increased in the Skeletal Muscle of Mdx Mice

Elevated levels of cytokines, including IL-1β, have been shown in the skeletal muscle tissue of *mdx* mice [10]. Nevertheless, the expression of inflammasome proteins, both in skeletal muscle and cardiac tissue, has not been investigated so far. Therefore, we quantified the inflammasome priming by detecting the expression of canonical inflammasome sensors, NLRP3, AIM2, NLRC4, and non-canonical caspase-11, in parallel with downstream signaling mediators, such as an apoptosis-associated speck-like protein containing a CARD (ASC), gasdermin D (GSDMD), caspase-1, and interleukins (IL-1β and IL-18) using Western blot analysis. We also detected the cleavage of the downstream mediators to demonstrate the presence of inflammasome activation (Figure 1 and Figure 2).

We did not observe any significant change in the expression of the inflammasome sensors in *mdx* mice at 3 and 10 months of age. However, enhanced priming of the inflammasome adaptor protein ASC and GSDMD was observed in the skeletal muscle at 3 and 10 months old. In addition to priming, the cleavage of IL-18 demonstrated inflammasome activation at the late stage (Figure 2A). Interestingly, a slight tendency for cleaved IL-1β elevation was also detected at an early stage and diminished at a late stage, indicating dynamic, time-dependent changes in the inflammasome activity (Figure 1A and Figure 2A).

In contrast to the moderate inflammation observed in the skeletal muscle, only slight, non-significant changes were found in the heart tissue of *mdx* mice (Figure 1B and Figure 2B). A tendency toward elevated expression of the different inflammasome proteins (e.g., GSDMD, ASC) was found without alteration in the levels of sensors, cytokines or other inflammasome components at an early stage. Interestingly, mild elevations in cleaved caspase-1 and GSDMD were found at a late stage of DMD in the heart tissue of *mdx* mice, indicating that a low-degree of inflammasome activity might be present in the heart, albeit no inflammasome priming or cytokine release was confirmed (Figure 2B).

Inflammasomes and related inflammatory mediators are expressed and activated in the cells and organs of the innate immune system of the monocyte/macrophage linage [13]. We detected only mild inflammation in the skeletal muscles and cardiac tissues, especially at a later stage, we examined the presence of innate immune cells in the tissue samples. Therefore, we examined the expression of leukocyte infiltration and immunohistochemistry was performed to identify CD68- and myeloperoxidase (MPO)-positive cells, using the general markers of monocyte–macrophage and granulocyte lineage, respectively (Figure 2C, Appendix A).

Focal leukocyte infiltration was detected in skeletal muscle at a late stage in the mouse *mdx* model (Figure 2C). Interestingly, leukocyte infiltration with CD68+ or MPO+ cells was observed in the myocardium similarly to the skeletal muscle, despite the lack of clear evidence for enhanced activity of the inflammasomes (Figure 2C, Appendix A).

### 2.2. Inflammasome Activity Was Enhanced in the Skeletal Muscle of DMD^mdx^ Rats without Inflammation in the Heart Muscle

Although the *mdx* mouse model is the most widely used for investigating DMD, new genetic models for DMD have also been developed, such as the DMD^mdx^ rat model [14]. It was previously shown that DMD*^mdx^* rats demonstrate the features of cardiomyopathy at 9 months old, such as reduced cardiac function and increased expression of pro-inflammatory mediators including IL-1β, suggesting that the DMD^mdx^ rat is a promising model to characterize DMD-associated inflammation and, eventually, its role in the progression of cardiomyopathy [15]. Thus, identical analysis was conducted on the skeletal muscle and heart tissue samples collected from DMD*^mdx^* rats. At an early stage of muscular damage, both the expression and cleavage of the cytokines (IL-1β, IL-18) were elevated significantly, indicating the priming and activation of inflammasome effectors (Figure 3A). The protein expression of GSDMD was tendentiously increased (Figure 3A). However, no cleavage of caspase-1 was observed at the same time point (Figure 3A). At the later stage of the disease, less pronounced inflammasome activity was found compared to the corresponding control; a mild priming of IL-18 and GSDMD was observed and the enhanced cleavage of cytokines (Figure 4A).

Surprisingly, inflammasome priming and activation was less prominent in the heart tissue compared to skeletal muscle in DMD^mdx^ rats. Non-specific, tendentious alterations in the protein expressions were observed at both time points, including increased cleavage of caspase-11 and priming of AIM2 at an early stage, while no changes were observed at a later stage (Figure 3B and Figure 4B).

When leukocyte infiltration was examined, focal leukocyte infiltration was found in both the skeletal muscle and heart samples at a late stage in DMD^mdx^ rats (Figure 4C and Appendix A).

### 2.3. Skeletal Muscle and Cardiac Fibrosis

Inflammatory activity is frequently associated with progressive fibrosis in the heart or skeletal muscle, in particular at a later stage, as was reported previously [15,16,17]. Thus, we have examined the presence of fibrosis in both heart and gastrocnemius tissue samples at a later stage of the disease. In line with the previous observations, we found that the cardiac fibrosis was markedly increased in both animal models (Figure 5). We demonstrated the presence of fibrosis in the skeletal muscle tissue of *mdx* mouse, but we did not observe significant alterations in the DMD^mdx^ rats (Figure 5).

## 3. Discussion

We have demonstrated altered inflammasome priming and activation in both the skeletal and heart muscle in one rat and one mouse model of DMD. It is important to emphasize that we compared the expression and activity of inflammasome signaling within one species, but not intercross between the two species. Overall, the early stage of muscle damage was associated with enhanced inflammasome activity, such as the increased expression of various inflammasome proteins as well as the enhanced cleavage of cytokines that was attenuated at a later stage of the disease in both models. Inflammasome priming and activation showed a distinct pattern between the two models, as DMD^mdx^ rats showed a marked increase in the expression and cleavage of cytokines. In contrast, the *mdx* mouse was characterized by the presence of slight priming rather than activation. Of note, significant differences were observed between the patterns of inflammatory activity in the skeletal muscle and heart tissue samples even in the same model. In general, the cardiac muscle was characterized by less pronounced inflammation, as we found no significant differences compared to the corresponding controls. However, leukocyte infiltration and cardiac fibrosis were still observed in both DMD models.

Loss of dystrophin in mammalian cells induces cell damage that triggers leukocyte infiltration and inflammation, consequently inflammatory signaling is an important player in the progression of DMD [1,18]. Interestingly, it has been shown that immunodeficiency and a lack of T, B, and NK cells did not influence fibrosis significantly; however, innate immune cells, such as monocytes, macrophages or granulocytes, might play a role in the maintenance of the pro-inflammatory environment [19]. It has been reported that the NLRP3 inflammasome and IL-1β might be significant players in the progression of the disease [9,10,20]. Elevation in the protein expression of various inflammasome components, including NLRP3, ASC, and pro-IL-1β, has been shown in the skeletal muscle of *mdx* mouse at an early stage [20,21]. Despite the occurrence of elevated protein expression, it was also reported that genetic disruption of ASC did not improve the outcome in the *mdx* mouse model raising the question as to whether inflammasomes are a potential target in DMD [21]. In a separate study, *mdx* mice crossed with NLRP3-knockout mice showed reduced muscle loss and higher global muscle force, indicating the critical and causative role of NLRP3 inflammasome activation in muscular damage [10]. A possible explanation for the apparent discrepancy between the referenced studies can be explained by the regulation of the inflammasome-independent pro-inflammatory processes through inflammasome components [12,22,23]. In our study, the protein expression of NLRP3 in the skeletal muscle tissue of *mdx* mice did not show a significant dysregulation. Albeit tendencies towards elevation in the levels of pro-caspase-1, GSDMD, ASC, and cleaved cytokines confirm the enhanced priming and activity of inflammasomes at an early stage, observed previously by another research group [10]. A similar pattern was found at a later time point in the skeletal muscle with significant IL-18 release, although inflammasome priming and activation was generally milder compared to the early time point.

Of note, we found the increased expression or activation of two of the inflammasome components, namely IL-18 and GSDMD, in the DMD models. IL-18 is recognized as a pro-fibrotic cytokine; and elevated levels of IL-18 are associated with cardiac fibrosis. Consequently, targeting IL-18 is a potential anti-fibrotic approach [24,25,26]. The level of IL-18 was increased in the skeletal muscle of *mdx* mice and DMD*^mdx^* rats, even at the late stage, suggesting that it may become a promising target in the reduction of skeletal muscle fibrosis and stiffness. In contrast, IL-18 expression did not show a similar pattern of changes in cardiac tissue, thereby its role in the progression of cardiac fibrosis may not be essential in DMD. The expression of GSDMD, a pore-forming protein that is generally responsible for pyroptosis, a pro-inflammatory type of programmed cell death was altered in the *mdx* mouse model. Previous studies have shown that the upregulation of GSDMD activity is linked to the loss of skeletal muscle tissue and myocardial ischemia-reperfusion injury [27,28]. In line with the previous findings, GSDMD showed mild elevation in the skeletal muscle in our study, suggesting that pyroptosis may contribute to muscle loss in DMD. Interestingly, GSDMD-mediated pyroptosis is associated with glucocorticoid-induced muscular atrophy via various mechanisms [29,30], which is in clear contrast to the clinical observation that glucocorticoids are beneficial in DMD patients. In fact, glucocorticoid-mediated protection was also reported along with other drugs [31]. Thus, further studies are warranted to explore and clarify the role of IL-18 and GSDMD in the progression of muscular atrophy and dysfunction in DMD.

The clinical outcome of DMD, as well as the response to pharmacological therapy, may vary among patients [32]. Our investigation revealed that inflammasome activity shows a distinct pattern in rodent models of DMD. The DMD*^mdx^* rat was characterized by significant cytokine release in the skeletal muscle without inflammasome priming, while the *mdx* mouse model showed the priming of ASC and GSDMD with no pronounced cytokine cleavage. These findings suggest that the genetic background and species differences are important factors in the inflammasome response in DMD [32,33]. The differences in the molecular pathology between the DMD models highlight the limitations of the current animal models used as drug testing platforms.

Inflammasomes, especially NLRP3, are believed to be key players in various cardiovascular events, such as acute myocardial infarction, myocarditis or pulmonary hypertension [34,35,36]. Moreover, non-NLRP3 inflammasomes, including NLRC4 and AIM2, contribute to the development of chronic heart failure [13,37]. Until now, there is a general lack of evidence on the role of inflammasomes in DMD-associated cardiomyopathy. Surprisingly, the heart tissue showed less prominent inflammasome priming or activation compared to the skeletal muscle in both DMD models. With respect to cardiomyopathy, only the DMD^mdx^ rats showed a significant reduction in the left ventricle ejection fraction (LVEF) at 9 months of age [15], while the *mdx* mice showed preserved LVEF at a comparable age. Collectively, inflammasome priming and activation may not be directly linked to the progression of DMD-associated cardiomyopathy. Nevertheless, further studies are warranted to clarify whether the inflammasome activation at the advanced age of animals and, consequently, at the end stage of the disease may play a role in DMD cardiomyopathy.

In contrast to the attenuated inflammasome activity, focal leukocyte infiltration and fibrosis were confirmed at later stages in both models. It has been previously shown that inflammasome activity is associated with enhanced fibrosis in various tissues [38]. In our study, we also demonstrated fibrosis in both the skeletal muscle and heart of DMD animals. Our findings are in line with previous reports that cardiac fibrosis develops at a later stage in *mdx* mice, such as at 7 or 10 months, while less or no fibrosis is observable at early time points including 6 or 8 weeks [16,17]. Furthermore, cardiac and skeletal muscle fibrosis has been detected at an earlier time point in DMD^mdx^ rats, suggesting that more prominent inflammatory processes might be present and act as an initiator or driver of progressive fibrosis [14,15]. Surprisingly, we did not demonstrate enhanced fibrosis in the skeletal muscle of DMD^mdx^ rats. Of note, other pro-fibrotic pathways, such as transforming growth factor beta, was shown to be a key player in DMD-associated fibrosis, suggesting that the detailed connections between inflammasome activity and fibrotic activity should be clarified in the future [39].

Similarly, focal CD68+ and MPO+ cell infiltration was observed in both models and tissue, indicating the presence of different leukocytes in the tissue samples at a later stage. Previous papers have reported the enhanced migration and presence of leukocytes (e.g., neutrophils, different phenotypes of macrophages) in tissue from DMD patients and animal models [18,39]. Furthermore, asynchronous degeneration and regeneration were demonstrated in DMD patients that shows a similar pattern to the observed focal leukocyte infiltration [39]. The infiltrating leukocyte populations have been shown to be heterogeneous based on their surface marker patterns and are also present at an early time point [39,40]. Interestingly, it was reported that the composition of infiltrating leukocyte populations can be distinct between the skeletal muscle and heart tissue, as well as between different muscle types, such as the diaphragm or the biceps femoris [41]. Our results accompanied with the data in the literature also indicates that the mechanisms leading to DMD-associated cardiomyopathy are distinct at some points from those causing skeletal muscle damage. Therefore, further research is needed to explore whether anti-inflammatory strategies can exert the same efficacy in slowing disease progression in DMD-associated cardiomyopathy.

There are certain acknowledged limitations. We performed experiments only on two well-established rodent models for DMD; however, large animal models of DMD, as well as human data are critical to confirm our findings. Furthermore, the different characteristics of human and rodent immune systems (such as the site of the hematopoiesis, differences between the subsets of leukocytes and receptor patterns, e.g., Toll-like receptors) should be taken into consideration when inflammatory reactions and anti-inflammatory therapies are translated into human clinical practice [42]. We used skeletal muscle samples collected from the gastrocnemius muscle that might show distinct inflammatory signalization compared to intercostal muscles and the diaphragm. The disturbances of these respiratory muscle functions play a fundamental role in DMD pathology.

## 4. Materials and Methods

### 4.1. Rodent Models for Duchenne Muscular Dystrophy and Tissue Collection

The male dystrophin-deficient DMD^mdx^ rats [14] originated from INSERM-CRTI UMR 1064 and were bred at the core facility in the laboratory for animal breeding and husbandry at the Medical University of Vienna, Austria. Genetic genotyping of the rats was performed using standard PCR assays, as described previously [14]. Male DMD^mdx^ (n = 7) and wild-type littermate control (n = 6) Sprague Dawley rats were used. DMD^mdx^ mutation is a deletion of 11 bp in exon 23 leading to a +1 frame shift and premature stop codon 81 bp after the mutation, according to Larcher et al. [14]. Male *mdx* (C57BL/10ScSn-Dmdmdx/J; n = 5–6) and wild-type littermate control (C57BL/10ScSnJ; n = 4–6) mice were used from Charles River Laboratories [43]. Mdx mutation is a nonsense point mutation (C-to-T transition) in exon 23 that aborts full-length dystrophin expression [43]. For the experiments, the animals were examined at 3 months (early stage) or 9–10 months (late stage), based on previous characterization studies reporting that cardiac phenotypes (e.g., function, fibrosis) can be markedly different at these time points [5,15,16]. The rats and mice were anesthetized by intraperitoneal injection of a mixture of xylazine (4 mg/kg; Bayer, Berlin, Germany) and ketamine (100 mg/kg; Dr E. Gräub AG, Bern, Switzerland), and the heart and skeletal muscle (gastrocnemius muscle) samples were removed. The samples were rinsed immediately in saline, blotted dry, frozen in liquid nitrogen, and kept at −80 °C until further processing. Another series of tissue samples was formalin-fixed, paraffin-embedded (FFPE) for histology.

### 4.2. Western Blot Analysis

In order to investigate whether inflammasome expression is altered at the protein level in the homogenates, Western blotting was performed, as described previously [13]. Frozen tissue samples from the left ventricle were homogenized in 1× radio immunoprecipitation assay buffer (RIPA; Cell Signaling Technology, Danvers, MA, USA) supplemented with 1× HALT protease and phosphatase inhibitor cocktail (Thermo Fisher Scientific, Waltham, MA, USA). Protein concentration was determined using the bicinchoninic acid assay kit (Thermo Fisher Scientific, Waltham, MA, USA). Equal amounts of protein from each sample were mixed with 1/4 volume of Laemmli buffer containing β-mercaptoethanol (Thermo Fisher Scientific, Waltham, MA, USA) and were loaded on 4–20% Tris-glycine sodium dodecyl sulfate-polyacrylamide gels (Bio-Rad, Hercules, CA, USA), and electrophoresed. The proteins were transferred onto a polyvinylidene difluoride membrane (Bio-Rad, Hercules, CA, USA) with the Trans-Blot^®^ Turbo™ Transfer System (Bio-Rad, Hercules, CA, USA). The membranes were blocked in 5% bovine serum albumin (Bio-Rad, Hercules, CA, USA) in Tris-buffered saline containing 0.05% Tween-20 (0.05% TBS-T; Sigma, St. Louis, MO, USA) for 2 h at room temperature, and then probed with primary antibodies overnight at 4 °C (primary antibodies: see Appendix A for antibodies and dilution). After three washes in TBS-T, the membranes were incubated with corresponding HRP-conjugated secondary antibodies (Cell Signaling, Danvers, MA, USA) for 2 h and washed in TBS-T. The signals were visualized using an enhanced chemiluminescence kit (Bio-Rad, Hercules, CA, USA) by Chemidoc XRS+. Image analysis was performed using Image Lab™ 6.0 software (Bio-Rad, Hercules, CA, USA).

### 4.3. Histology and Immunohistochemistry

The histology and immunohistochemistry were performed as previously described [13]. After routine FFPE specimen processing, 4 µm thick tissue sections were prepared. For Sirius red staining, the slides were stained with 0.0125% picrosirius red for 1 h, then washed with 1% acetic acid. The extent of the fibrosis in the cardiac and skeletal muscle tissue was analyzed and quantified using the ImageJ ver1.53t software.

For the immunohistochemistry, deparaffinized sections underwent antigen retrieval (pH = 6 citrate buffer, at 95 °C for 15 min). After blocking the endogenous peroxidase activity (3% H2O2 solution in PBS), sections were blocked in the appropriate sera (2.5% goat, bovine serum or milk diluted in PBS). The sections were incubated with the primary antibodies (see Appendix A for antibodies and dilution) in diluted blocking solution overnight at 4 °C. After primary antibody incubation, the sections were washed three times in PBS and incubated for an hour with peroxidase polymer conjugated (ImmPRESS reagents, Vector Laboratories, Burlingame, CA, USA) anti-rabbit IgG or anti-mouse IgG. The secondary antibodies were washed three times for 10 min and the specific signal was developed with diaminobenzidine (ImmPACT DAB EqV Peroxidase (HRP) substrate, Vector Laboratories, Burlingame, CA, USA). Images were acquired using a bright field microscope (Leica DM3000, Leica Microsystems, Wetzlar, Germany).

### 4.4. Data Analysis

All data are expressed as mean ± SEM. Comparisons of the two groups were performed using the unpaired Student’s *t*-test. Moreover, *p* < 0.05 was considered statistically significant. Statistical analysis was performed using GraphPad Prism 8 (GraphPad Software Inc., San Diego, CA, USA).

## 5. Conclusions

Signs of inflammasome priming and activation, such as increased expression of the adaptor protein or the enhanced cleavage of cytokines may be present in the skeletal muscle at an early stage of the disease in the *mdx* mouse and DMD*^mdx^* rat model. In agreement with clinical observations, this inflammatory activity was reduced over time in both models. In addition, the observed inflammatory response in the skeletal muscle and cardiac tissue has distinct molecular patterns since inflammasome priming and activation are less prominent in the heart compared to the skeletal muscle. These results highlight possible differences in the pathomechanism of skeletal damage and the progression of cardiomyopathy in DMD, suggesting the need for novel targeted therapeutic approaches.

## Figures and Tables

**Figure 1 ijms-24-08497-f001:**
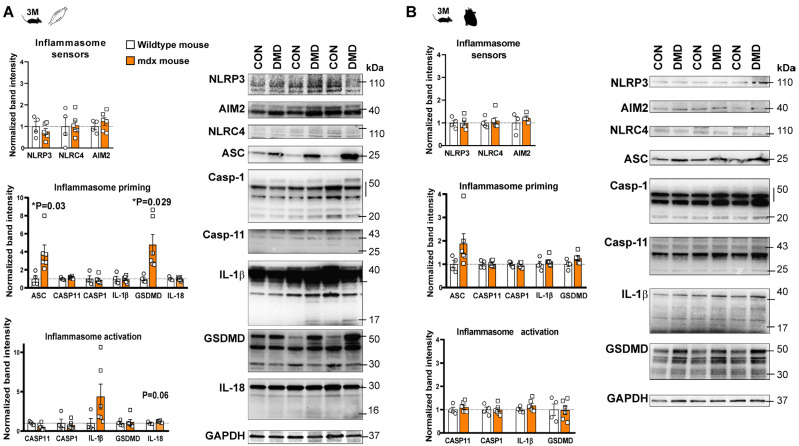
Assessment of inflammasome priming and activity in skeletal and cardiac muscle samples of 3-month old *mdx* mice. Western blot analysis of the inflammasome sensors (NLRP3, AIM2, NLRC4) and inflammasome activation related proteins (ASC, caspase-1, IL-1β, IL-18, and gasdermin D) in skeletal muscle (**A**); and left ventricular tissue (**B**) at the early stages of muscular dystrophy. * *p* < 0.05 vs. wild-type littermates, Student’s *t*-test; n = 4–6.

**Figure 2 ijms-24-08497-f002:**
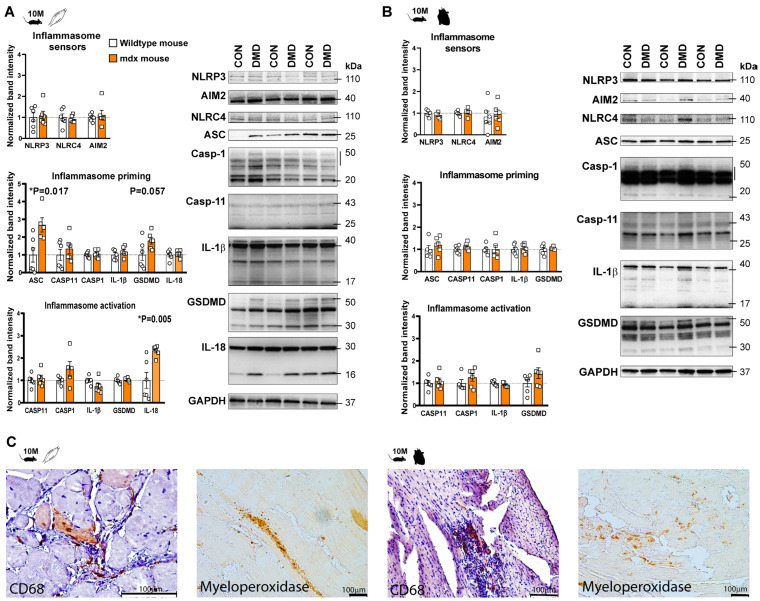
Assessment of inflammasome priming and activity in skeletal and cardiac muscle samples of 10-month old *mdx* mice. Western blot analysis of the inflammasome sensors (NLRP3, AIM2, NLRC4) and inflammasome activation related proteins (ASC, caspase-1, IL-1β, IL-18, and gasdermin D) in skeletal muscle (**A**); and left ventricular tissue (**B**) at the late stages of muscular dystrophy. * *p* < 0.05 vs. wild-type littermates, Student’s *t*-test; n = 5–6. (**C**) Representative images from immunohistochemical detection of CD68 and myeloperoxidase proteins in skeletal and cardiac muscle samples. Hematoxylin was used for the counterstain. Scale bar: 100 μm.

**Figure 3 ijms-24-08497-f003:**
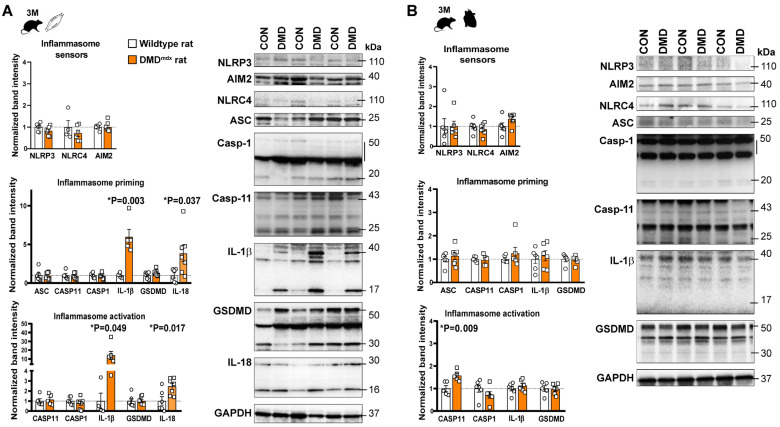
Assessment of inflammasome priming and activity in skeletal and cardiac muscle samples of 3-month old DMD^mdx^ rats. Western blot analysis of the inflammasome sensors (NLRP3, AIM2, NLRC4) and inflammasome activation related proteins (ASC, caspase-1, IL-1β, IL-18, and gasdermin D) in skeletal muscle (**A**); and left ventricular tissue (**B**) at the early stages of muscular dystrophy. * *p* < 0.05 vs. wild-type littermates, Student’s *t*-test; n = 5–7.

**Figure 4 ijms-24-08497-f004:**
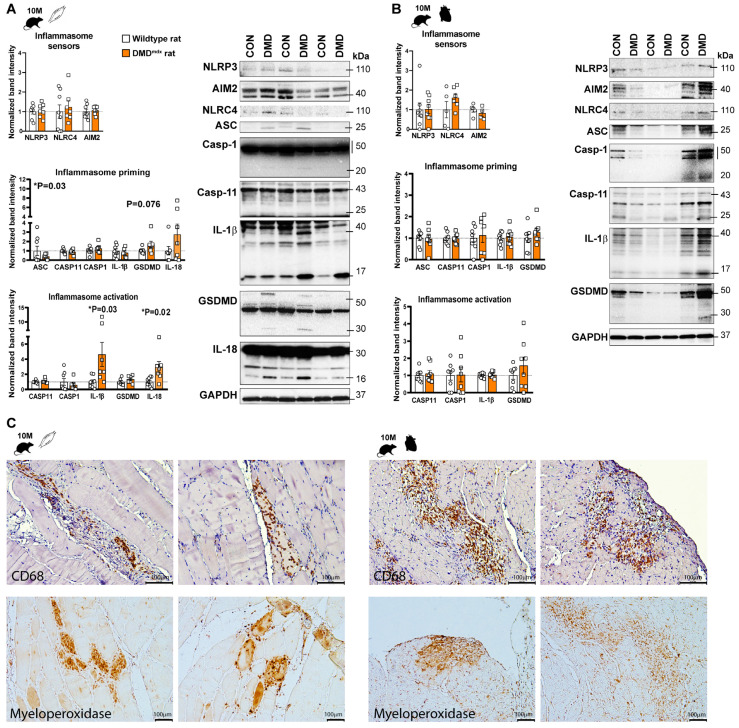
Assessment of inflammasome priming and activity in skeletal and cardiac muscle samples of 10-month old DMD^mdx^ rats. Western blot analysis of the inflammasome sensors (NLRP3, AIM2, NLRC4) and inflammasome activation related proteins (ASC, caspase-1, IL-1β, IL-18, and gasdermin D) in skeletal muscle (**A**); and left ventricular tissue (**B**) at the late stages of muscular dystrophy. * *p* < 0.05 vs. wild-type littermates, Student’s *t*-test; n = 6–8. (**C**) Representative images from immunohistochemical detection of CD68 and myeloperoxidase proteins in skeletal and cardiac muscle samples. Hematoxylin was used for the counterstain. Scale bar: 100 μm.

**Figure 5 ijms-24-08497-f005:**
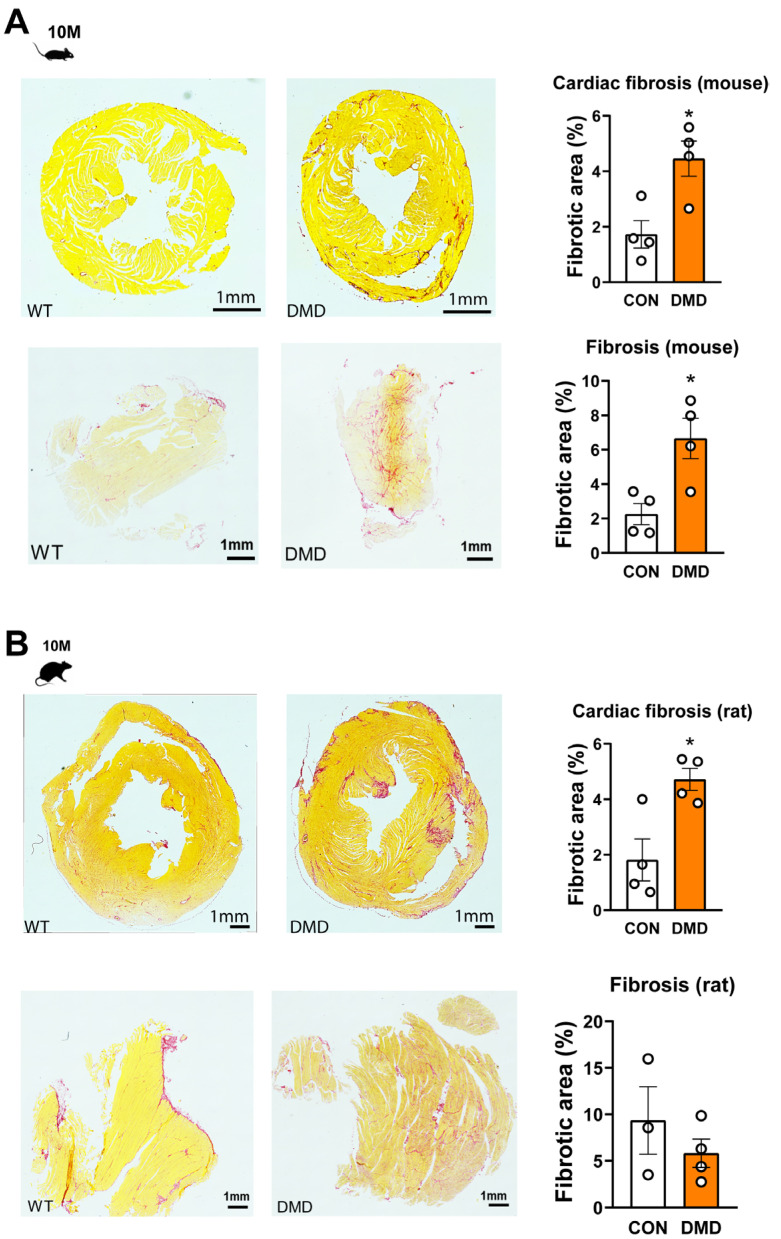
Assessment of cardiac and skeletal muscle fibrosis in DMD rodent models using Sirius red staining. Representative images and estimated fibrotic area in the cardiac tissue and gastrocnemius muscle samples from *mdx* mouse (**A**); and DMD^mdx^ rats (**B**). The fibrotic area was normalized to the total area. Scale bar = 1 mm. * *p* < 0.05 vs control, Student’s *t*-test, n = 3–4.

## Data Availability

The datasets used and/or analyzed are available from the corresponding author upon request.

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
