# Peer review of "Inflammasome Activity in the Skeletal Muscle and Heart of Rodent Models for Duchenne Muscular Dystrophy"

_ijms, 2023, doi:10.3390/ijms24108497_

Round 1
Reviewer 1 Report
The manuscript of Onodi et al. focuses on DMD, a disease affecting both skeletal muscles and the heart, and whose pathogenesis includes activation of the inflammatory response. On this trail, the authors here investigate ‘inflammasome priming and activation' in both the heart and the skeletal muscle, in mouse and rat preclinical DMD models.
In the present form, the manuscript assesses the protein content of inflammation markers in the heart vs skeletal muscle, at early and late disease stages.
Major comments:
1) In the opinion of this reviewer, it would be interesting to better correlate muscle/heart protein content of inflammatory markers with the respective muscle/heart phenotype. Thus, histological characterization of rat and mouse muscle and heart sections (i.e. analysis of cardiomyocyte/myocyte death, use of a large panel of inflammatory cells, state of fibrosis, cell atrophy), at early and advanced disease stage, need to accompany biochemical analyses.
2) Focusing on the heart, is the inflammasome priming and activation similar in the RV, IVS and LV?
3) In murine DMD models, utrophin is expressed instead of dystrophin. Did the authors evaluate the possible effects of utrophin expression in affecting inflammatory response? Utrophin expression may lead to results of difficult transability to the clinical context. This aspect should be commented in the Limitation section.
4) Is the profile of inflammatory cells different in the heart vs the skeletal muscle?
5) Did the authors observe differences in the inflammasome priming and activation in different muscle types (i.e. gastrocnemious vs. soleus vs. EDL)?
Author Response
The manuscript of Onodi et al. focuses on DMD, a disease affecting both skeletal muscles and the heart, and whose pathogenesis includes activation of the inflammatory response. On this trail, the authors here investigate ‘inflammasome priming and activation' in both the heart and the skeletal muscle, in mouse and rat preclinical DMD models.
In the present form, the manuscript assesses the protein content of inflammation markers in the heart vs skeletal muscle, at early and late disease stages.
We thank the Reviewer for the valuable comments. We have answered the concerns one-by-one.
Major comments:
1) In the opinion of this reviewer, it would be interesting to better correlate muscle/heart protein content of inflammatory markers with the respective muscle/heart phenotype. Thus, histological characterization of rat and mouse muscle and heart sections (i.e. analysis of cardiomyocyte/myocyte death, use of a large panel of inflammatory cells, state of fibrosis, cell atrophy), at early and advanced disease stage, need to accompany biochemical analyses.
We fully agree with the Reviewer, that detailed immunohistological and biochemical characterization of the heart and skeletal muscle samples certainly improve the quality of our paper and also necessary to clarify the role of inflammatory response in the pathomechanism of DMD. We have included the missing dataset on skeletal muscle fibrosis (Fig.5). However, we focused on investigating the inflammasome activity in the heart/skeletal muscle of the rodent models, as there are some papers reporting similar data. Thus, we have included a detailed discussion on previous results as the following:
“In contrast to the attenuated inflammasome activity, focal leukocyte infiltration and fibrosis were confirmed at later stages in both models. It has been previously shown that inflammasome activity is associated with enhanced fibrosis in various tis-sues (38). In our study, we also demonstrated the fibrosis in both the skeletal muscle and heart of DMD animals. Our findings were in line with previous reports that cardiac fibrosis develops at a later stage in mdx mice such as 7 or 10 months, while less or no fibrosis is observable at early time points including 6 or 8 weeks (16,17). Further-more, cardiac and skeletal muscle fibrosis has been detected at earlier time point in DMDmdx rats suggesting that more prominent inflammatory processes might be present and act as an initiator or driver of progressive fibrosis (14,15). Surprisingly, we did not demonstrate enhanced fibrosis in the skeletal muscle of DMDmdx rats. Of note, other pro-fibrotic pathways such as transforming growth factor beta was shown to be a key player in DMD-associated fibrosis suggesting that the detailed connections between inflammasome activity and fibrotic activity should be clarified in the future (39).
Similarly, focal CD68+ and MPO+ cell infiltration was observed in both models and tissue, indicating the presence of different leukocytes in the tissue samples at a later stage. Previous papers have reported the enhanced migration and presence of leukocytes (e.g. neutrophils, different phenotypes of macrophages) in tissue from DMD patients and animal models (18,39). Furthermore, asynchronous degeneration and regeneration were demonstrated in DMD patients that shows similar pattern to the observed focal leukocyte infiltration (39). The infiltrating leukocyte populations have been shown to be heterogeneous based on their surface marker patterns, and are also present at an early time point (39,40). Interestingly, it was reported that the composition of infiltrating leukocyte populations can be distinct between the skeletal muscle and heart tissue as well as between different muscles types such as the diaphragm or the biceps femoris (41). Our results ac-companied with the literature data also indicates that the mechanisms leading to DMD-associated cardiomyopathy are distinct at some points from the ones causing skeletal muscle damage. Therefore, further research is needed to explore whether anti-inflammatory strategies can exert the same efficacy in slowing disease progression in DMD-associated cardiomyopathy.”
2) Focusing on the heart, is the inflammasome priming and activation similar in the RV, IVS and LV?
We are grateful to the Reviewer for this comment. In our measurements (for Western blots), we used LV cardiac tissue samples. A previous clinical study by Mehmood et al (PMID: 26210092) demonstrated that even in patients with severe LV dysfunction, RVEF was relatively preserved, suggesting the different between RV and LV pathology in DMD. Nevertheless, whether the inflammasome priming and activation play role in this phenomenon has not been studied yet. Our data show that fibrosis present both in RV and LV, and macrophages were also present in both RV and LV tissue. However, whether inflammasome priming and activation similar or different in the RV, IVS and LV further study needs to clarify this issue.
3) In murine DMD models, utrophin is expressed instead of dystrophin. Did the authors evaluate the possible effects of utrophin expression in affecting inflammatory response? Utrophin expression may lead to results of difficult transability to the clinical context. This aspect should be commented in the Limitation section.
We are thankful to the Reviewer for drawing our attention to this point. It has been also shown that DMD and Becker muscular dystrophy patients express abnormally high levels of utrophin (PMID: 9364465). Utrophin may have a function similar to that of dystrophin, and compensate to some extent for dystrophin deficiency in DMD. Similar to the clinical context, we observed in LV tissue samples of Dmdmdx (PMID: 33619211) an increase of utropin expression compared to wildtype rats. Indeed, there is evidence that utrophin compensates, although insufficiently, for the effects of dystrophin loss with regard to inflammation and fibrosis of both quadriceps and diaphragm muscles in mdx mice (PMID: 17889902). However, our models at least in respect to utrophin expression are close to resemble the clinical context of DMD pathology.
4) Is the profile of inflammatory cells different in the heart vs the skeletal muscle?
This is an important question raised by the Reviewer. In our present study, we did not characterize the immune cells in the heart or skeletal tissue samples. However, we observed focal infiltration of CD68+ and MPO+ cells in both heart and skeletal muscle tissue (see Fig.2C and Fig4.C) and this finding has been further discussed in the revised version of the MS (Discussion).
“Similarly, focal CD68+ and MPO+ cell infiltration was observed in both models and tissue, indicating the presence of different leukocytes in the tissue samples at a later stage. Previous papers have reported the enhanced migration and presence of leukocytes (e.g. neutrophils, different phenotypes of macrophages) in tissue from DMD patients and animal models (18,39). Furthermore, asynchronous degeneration and regeneration were demonstrated in DMD patients that shows similar pattern to the observed focal leukocyte infiltration (39). The infiltrating leukocyte populations have been shown to be heterogeneous based on their surface marker patterns, and are also present at an early time point (39,40). Interestingly, it was reported that the composition of infiltrating leukocyte populations can be distinct between the skeletal muscle and heart tissue as well as between different muscles types such as the diaphragm or the biceps femoris (41). Our results ac-companied with the literature data also indicates that the mechanisms leading to DMD-associated cardiomyopathy are distinct at some points from the ones causing skeletal muscle damage. Therefore, further research is needed to explore whether anti-inflammatory strategies can exert the same efficacy in slowing disease progression in DMD-associated cardiomyopathy.”
5) Did the authors observe differences in the inflammasome priming and activation in different muscle types (i.e. gastrocnemious vs. soleus vs. EDL)?
We used skeletal muscle samples were collected from gastrocnemius muscle that might show distinct in the inflammasome priming and activation compared to intercostal muscles, soleus muscle and diaphragm. The disturbances of these respiratory muscles function may play a fundamental role in DMD pathology (see in the Limitations).
Reviewer 2 Report
Review of ijms 2334704
Inflammasome activity in the skeletal muscle and heart of rodent models for Duchenne muscular dystrophy (DMD)
This paper examines levels of inflammatory pathway biomarkers in skeletal (gastrocnemius) and cardiac muscle in mice and rat models of Duchenne muscular dystrophy, both early (3 months of age) and late (9-10months of age) in the disease process, comparing with same age controls. Those inflammatory markers which were raised in comparison with controls, tended to be so in skeletal rather than cardiac muscle (or with different ones and more mildly raised in cardiac muscle), and more prominently so in younger rather than older DMD-model animals.
The study seems well executed, and generally well-written, and the results add support to the principal of early use of steroid therapy in DMD in boys. The results also raise the possibility that addressing inflammation in cardiac muscle in DMD may require different therapeutic agents from those directed against inflammation in skeletal muscle.
Comments
1. One scientific comment is that the authors should include mention of the specific mutations involved (and their nature in terms of type of mutation) in the mdx mouse, and in the DMD mdx rat model, in case any element of inflammation is dependent on the nature of the altered dystrophin protein that is produced, and hence might differ in humans between boys with different types of mutation.
2. Where abbreviations or gene symbols are used repeatedly, it would help if their name could be written in full at first mention in the paper (eg. for ASC, GSDMD). For others (and including these most-used ones written out in the text) it would help to have a glossary at the end (or as a supplementary file if more appropriate).
Other comments are mostly directed at the written language.
3. Intro. Line 44 : ‘regime’ or ‘regimen’ rather than ‘regiment’
4. Intro. Line 50 : ‘’primarily’ rather than ‘primary’
5. Intro lines 51 : In mentioning ‘combined therapy’, the authors should clarify that this refers to (presumably) the 2 components of anti-inflammatory measures and dystrophin replacement/substitution, as the latter must be mentioned in talking about therapy.
6. Intro Line 60-61 : There needs to be punctuation here. Eg. ‘…by several mechanisms; a process that is referred to….’
7. Intro. Line 63 : ‘trigger’…’lead’ rather than: ‘triggers’…leading’…
8. Results Line 114 : ‘…that the DMDmdx rat…’ rather than ‘…that DMDmdx rats…’
9. Results Line 120 : ‘tendentious’ . Is this the correct word here ? If so it would be ‘tendentiously’ , but in this context, the authors may wish to check the meaning of that word, and select an alternative if appropriate.
10. Discussion Line 159 : ‘..demonstrated that there is…’ OR ‘…demonstrated that there is evidence for…’ OR ‘…demonstrated altered inflammasome…’ rather than ‘…demonstrated that altered…’
11. Discussion Line 174 : ‘…might be active…’ OR ‘…might play a role…’ OR ‘..might be players…’ rather than ‘…might play…’
12. Discussion Line 191 : ‘…at the later time point…’ rather than ‘…at late time point…’
13. Discussion Line 200 : ‘does not show a similar’ OR ‘did not….’ rather than ‘was not…’
14. Discussion Line 201 : ‘may not be essential’ rather than ‘may not essential’
15. Discussion Line 203 : ‘did show alteration’ rather than ‘was shown alteration’
16. Discussion Line 210 : ‘In fact…’ might be better than ‘In contrary….’
17: Discussion Line 219-220 : ‘These findings suggest….’ This sentence needs re-phrasing. Eg. ‘These findings suggest that genetic background and species differences are important factors in inflammasome response in different species models of DMD’.
18. Discussion Line 228 : ‘heart’ rather than ‘hearts’
19. Discussion Line 229 : ‘In considering cardiomyopathy…’ OR ‘In consideration of….’ OR ‘With respect to…’ rather than : ‘In highlight of….’.
20. Discussion Line 235 : ‘..play a role…’ OR ‘…have a role…’ rather than ‘..play role…’
21. Discussion Line 236 : ‘…at the late stage..’ OR ‘…at the later stage…’ rather than : ‘…at late stage…’
22. Discussion Line 237 : ‘…mechanisms… …cardiomyopathy are distinct…’ rather than …’is distinct…’.
23. M & M Line 246-247 : See Main Point 1 above.
24 M & M Line 251 : I would suggest to Delete ‘in order to remove’ and replace with ‘and’
25. M & M Line 288 : Please include PBS in the Glossary ( as well as the many other abbreviations).
Author Response
This paper examines levels of inflammatory pathway biomarkers in skeletal (gastrocnemius) and cardiac muscle in mice and rat models of Duchenne muscular dystrophy, both early (3 months of age) and late (9-10months of age) in the disease process, comparing with same age controls. Those inflammatory markers which were raised in comparison with controls, tended to be so in skeletal rather than cardiac muscle (or with different ones and more mildly raised in cardiac muscle), and more prominently so in younger rather than older DMD-model animals.
The study seems well executed, and generally well-written, and the results add support to the principal of early use of steroid therapy in DMD in boys. The results also raise the possibility that addressing inflammation in cardiac muscle in DMD may require different therapeutic agents from those directed against inflammation in skeletal muscle.
We thank the Reviewer for the valuable comments and corrections.
Comments
- One scientific comment is that the authors should include mention of the specific mutations involved (and their nature in terms of type of mutation) in the mdx mouse, and in the DMD mdx rat model, in case any element of inflammation is dependent on the nature of the altered dystrophin protein that is produced, and hence might differ in humans between boys with different types of mutation.
The most widely used animal model for DMD research is the mdx mouse. Mdx mutation is a nonsense point mutation (C-to-T transition) in exon 23 that aborted full-length dystrophin expression (PMID: 6583703). This is a naturally occurring, spontaneous mutation in inbred C57/BL10 mice.
DMDmdx mutation in rats is a deletion of 11 bp in exon 23 by transcription activator-like effector nucleases (TALENs) leading to a +1 frame shift and premature stop codon 81 bp after the mutation according to Larcher et al (PMID: 25310701).
- Where abbreviations or gene symbols are used repeatedly, it would help if their name could be written in full at first mention in the paper (eg. for ASC, GSDMD). For others (and including these most-used ones written out in the text) it would help to have a glossary at the end (or as a supplementary file if more appropriate).
We thank to Reviewer for highlighting these issues. We have included the details and the abbreviations list in the revised form of our MS and Supplemental Material.
“DMD: Duchenne muscular dystrophy; NF-κB: nuclear factor kappa B; IL-1β: interleukin 1 beta; NLRP3: NLR family, pyrin domain containing 3; AIM2: absent in melanoma 2; NLRC4: NLR family CARD domain-containing protein 4; TNFα: tumor necrosis factor alpha; DAMP, PAMP: danger- or pathogen-associated molecular patterns; IL-18: interleukin-18; ASC: Apoptosis-associated speck-like protein containing a CARD; GSDMD: gasdermin D; MPO: myeloperoxidase; PBS: phosphate buffer saline”
Other comments are mostly directed at the written language.
3. Intro. Line 44 : ‘regime’ or ‘regimen’ rather than ‘regiment’
- Intro. Line 50 : ‘’primarily’ rather than ‘primary’
- Intro lines 51 : In mentioning ‘combined therapy’, the authors should clarify that this refers to (presumably) the 2 components of anti-inflammatory measures and dystrophin replacement/substitution, as the latter must be mentioned in talking about therapy.
- Intro Line 60-61 : There needs to be punctuation here. Eg. ‘…by several mechanisms; a process that is referred to….’
- Intro. Line 63 : ‘trigger’…’lead’ rather than: ‘triggers’…leading’…
- Results Line 114 : ‘…that the DMDmdxrat…’ rather than ‘…that DMDmdxrats…’
- Results Line 120 : ‘tendentious’ . Is this the correct word here ? If so it would be ‘tendentiously’ , but in this context, the authors may wish to check the meaning of that word, and select an alternative if appropriate.
- Discussion Line 159 : ‘..demonstrated that there is…’ OR ‘…demonstrated that there is evidence for…’ OR ‘…demonstrated altered inflammasome…’ rather than ‘…demonstrated that altered…’
- Discussion Line 174 : ‘…might be active…’ OR ‘…might play a role…’ OR ‘..might be players…’ rather than ‘…might play…’
- Discussion Line 191 : ‘…at the later time point…’ rather than ‘…at late time point…’
- Discussion Line 200 : ‘does not show a similar’ OR ‘did not….’ rather than ‘was not…’
- Discussion Line 201 : ‘may not be essential’ rather than ‘may not essential’
- Discussion Line 203 : ‘did show alteration’ rather than ‘was shown alteration’
- Discussion Line 210 : ‘In fact…’ might be better than ‘In contrary….’
17: Discussion Line 219-220 : ‘These findings suggest….’ This sentence needs re-phrasing. Eg. ‘These findings suggest that genetic background and species differences are important factors in inflammasome response in different species models of DMD’.
- Discussion Line 228 : ‘heart’ rather than ‘hearts’
- Discussion Line 229 : ‘In considering cardiomyopathy…’ OR ‘In consideration of….’ OR ‘With respect to…’ rather than : ‘In highlight of….’.
- Discussion Line 235 : ‘..play a role…’ OR ‘…have a role…’ rather than ‘..play role…’
- Discussion Line 236 : ‘…at the late stage..’ OR ‘…at the later stage…’ rather than : ‘…at late stage…’
- Discussion Line 237 : ‘…mechanisms… …cardiomyopathy are distinct…’ rather than …’is distinct…’.
- M & M Line 246-247 : See Main Point 1 above.
24 M & M Line 251 : I would suggest to Delete ‘in order to remove’ and replace with ‘and’
- M & M Line 288 : Please include PBS in the Glossary ( as well as the many other abbreviations).
We are really grateful to the Reviewer for correcting these mistakes. We have corrected all of them in the revised form of our MS.
Reviewer 3 Report
The paper by Onodi et al, investigated makers of inflammation in mdx mice and rat skeletal and cardiac muscle, which is novel. However, I have several concerns with the manuscript. In general, the study is very descriptive and does not provide any evidence inflammation is driving DMD pathophysiology. That said, essentially no changes are observed in cardiac muscle and only minor to moderate changes are seen in skeletal muscle, suggesting that inflammation seen in rodents may not be translatable to DMD patients. That said, it is difficult to say that the few changes observed in skeletal muscle are actually related to inflammasome sensing, priming and activation. The results are also hard to follow because description of the data jumps around and there seems to be pieces missing (e.g., like only having one time point from some assessments but not others). The authors also need to justify what is early and late stages of disease progression as it relates to the mdx mouse AND rat. Lastly, the authors need to ensure they are referencing previous data, especially when making broad statements.
Line 24; please add in “a rodent models” of DMD”.
Abstract; please add in some data and p-values to support the results you described.
Line 43; add a references after “…..muscle tissue.”
Line 48; “in DMD”…. Please clarify when using this expression that you mean rodent model of DMD or patients with DMD. Please adjust the manuscript accordingly.
Lines 59-64; add in references.
Line 69; add in statement informing reader what these models are (e.g., mdx mouse and rat).
Lines 242-254; be consistent with superscripting mdx after DMD (for rats).
Line 249; please justify ages with specific rationale and references.
Lines 306-307; what characteristics? Also, please add in references.
Lines 309-311; add in references.
Results; why were no assessments done to compare the effect of age? Essentially, a measure of disease progression via the markers of inflammation.
Results; it is very confusing that Figure 2C is not discussed until section 2.3. Why is this data not discussed with the other mouse data? Where is the quantification for Figures 4C and 4C. Why is there only “late” stage data for these Figures?
Figures; in general, you list p value and also use significant symbol *. Typically, you use symbols in graphs and list p value in texts. Just an observation, this is fine as is.
Figures, blots; for many of the blots images it is unclear which band is the actual protein of interest. Please make sure this is clear.
Figure 3; inflammasome activation, please fix figure so line breaks don’t cut off top of bars (see IL-18). This makes the data hard to visualize and interpret.
Figure 5; do you have data from the “early” time point? Any data for the gastrocnemius?
Line 159; what do you mean by altered? The majority of the measurements you analyzed did not change. I did not see a single sensor, primer or activator differ between cardiac muscle in controls and DMD animals. Essentially, there was no quantifiable evidence of inflammation in cardiac muscle and some minor to moderate changes in skeletal muscle. None of which could be linked to age or disease progression.
Lines 170-172; add in references.
Line 180; be consistent with using italics for the mdx.
Conclusions; your data only shows a “consistent” increase in content of IL-1beta and IL-18 in gastrocnemius muscle of mdx mice and rats. It does not provide any actual evidence of inflammasome priming and activation. Moreover, you did not run any statistics over time (age effects) and therefore you can only say there was no difference from age-matched control. Your data also does not highlight any “differences in the pathomechanisms of skeletal damage and the progression of cardiomyopathy in DMD” because you did not measure any marker of damage and regardless of changes in inflammatory markers, both tissues were fibrotic.
Author Response
The paper by Onodi et al, investigated makers of inflammation in mdx mice and rat skeletal and cardiac muscle, which is novel. However, I have several concerns with the manuscript. In general, the study is very descriptive and does not provide any evidence inflammation is driving DMD pathophysiology. That said, essentially no changes are observed in cardiac muscle and only minor to moderate changes are seen in skeletal muscle, suggesting that inflammation seen in rodents may not be translatable to DMD patients. That said, it is difficult to say that the few changes observed in skeletal muscle are actually related to inflammasome sensing, priming and activation. The results are also hard to follow because description of the data jumps around and there seems to be pieces missing (e.g., like only having one time point from some assessments but not others). The authors also need to justify what is early and late stages of disease progression as it relates to the mdx mouse AND rat. Lastly, the authors need to ensure they are referencing previous data, especially when making broad statements.
We appreciate the Reviewer for the constructive comments. We have answered all the concerns one-by-one and improved the manuscript according to them.
Line 24; please add in “a rodent models” of DMD”.
Abstract; please add in some data and p-values to support the results you described.
Line 43; add a references after “…..muscle tissue.”
Line 48; “in DMD”…. Please clarify when using this expression that you mean rodent model of DMD or patients with DMD. Please adjust the manuscript accordingly.
Lines 59-64; add in references.
Line 69; add in statement informing reader what these models are (e.g., mdx mouse and rat).
Lines 242-254; be consistent with superscripting mdx after DMD (for rats).
Line 249; please justify ages with specific rationale and references.
Lines 306-307; what characteristics? Also, please add in references.
Lines 309-311; add in references.
We are grateful to the Reviewer for these comments. All the point has been changed and/or corrected in the revised version of the MS.
Results; why were no assessments done to compare the effect of age? Essentially, a measure of disease progression via the markers of inflammation.
That is an interesting question raised by the Reviewer. The aim of the study was to clarify whether the inflammasome priming and activation in cardiac and skeletal muscle tissue show different between age (3 and 10 months old) and sex matched control (wildtype) vs DMD animals. We need to emphasize that altered inflammasome activity might be associated with aging (PMID: 28092664, 35525426) that may complicate the evaluation of that kind of comparisons theoretically. Thus, we decided to compare DMD animals to their age matched controls, and we have only assessed the inflammasome priming and activation accordingly. As we observed increased inflammasome activity at an early time point, while less differences were confirmed at the late one, our data suggests that inflammasome activity is altered rather in the early than in the late phase of DMD muscular pathology.
Results; it is very confusing that Figure 2C is not discussed until section 2.3. Why is this data not discussed with the other mouse data? Where is the quantification for Figures 4C and 4C. Why is there only “late” stage data for these Figures?
We thank the Reviewer for highlighting these issues. We have modified the section according to the request of the Reviewer.
We agree with the Reviewer that quantification of leukocyte infiltration would improve the quality of our paper. Unfortunately, we have only very small amount of samples from the series included in this present paper that limits our opportunity for evaluation in a reliable way. Nevertheless, DMD rodent models are frequently used at 6-12 weeks (early time point); thus, previous literature data is available on leukocyte infiltration in DMD rodent models as well as human patients. To address the question of Reviewer, we have discussed the presence of leukocyte infiltration in skeletal muscle and heart extensively as the following:
“Similarly, focal CD68+ and MPO+ cell infiltration was observed in both models and tissue, indicating the presence of different leukocytes in the tissue samples at a later stage. Previous papers have reported the enhanced migration and presence of leukocytes (e.g. neutrophils, different phenotypes of macrophages) in tissue from DMD patients and animal models (18,39). Furthermore, asynchronous degeneration and regeneration were demonstrated in DMD patients that shows similar pattern to the observed focal leukocyte infiltration (39). The infiltrating leukocyte populations have been shown to be heterogeneous based on their surface marker patterns, and are also present at an early time point (39,40). Interestingly, it was reported that the composition of infiltrating leukocyte populations can be distinct between the skeletal muscle and heart tissue as well as between different muscles types such as the diaphragm or the biceps femoris (41). Our results ac-companied with the literature data also indicates that the mechanisms leading to DMD-associated cardiomyopathy are distinct at some points from the ones causing skeletal muscle damage. Therefore, further research is needed to explore whether anti-inflammatory strategies can exert the same efficacy in slowing disease progression in DMD-associated cardiomyopathy.”
Figures; in general, you list p value and also use significant symbol *. Typically, you use symbols in graphs and list p value in texts. Just an observation, this is fine as is.
Figures, blots; for many of the blots images it is unclear which band is the actual protein of interest. Please make sure this is clear.
Figure 3; inflammasome activation, please fix figure so line breaks don’t cut off top of bars (see IL-18). This makes the data hard to visualize and interpret.
We are grateful for highlighting the issues of figures. We have corrected the figures in order to improve the visualization of our data.
Figure 5; do you have data from the “early” time point? Any data for the gastrocnemius?
We have included the evaluation of the fibrosis in gastrocnemius muscle. Due to technical reasons, we have no available samples for histology at early time points to assess cardiac fibrosis. However, fibrotic changes of the skeletal muscle and heart in DMD models were examined at the relevant time points according to the literature. To address this issue, we have discussed these previous findings as the following:
“In contrast to the attenuated inflammasome activity, focal leukocyte infiltration and fibrosis were confirmed at later stages in both models. It has been previously shown that inflammasome activity is associated with enhanced fibrosis in various tis-sues (38). In our study, we also demonstrated the fibrosis in both the skeletal muscle and heart of DMD animals. Our findings were in line with previous reports that cardiac fibrosis develops at a later stage in mdx mice such as 7 or 10 months, while less or no fibrosis is observable at early time points including 6 or 8 weeks (16,17). Further-more, cardiac and skeletal muscle fibrosis has been detected at earlier time point in DMDmdx rats suggesting that more prominent inflammatory processes might be present and act as an initiator or driver of progressive fibrosis (14,15). Surprisingly, we did not demonstrate enhanced fibrosis in the skeletal muscle of DMDmdx rats. Of note, other pro-fibrotic pathways such as transforming growth factor beta was shown to be a key player in DMD-associated fibrosis suggesting that the detailed connections between inflammasome activity and fibrotic activity should be clarified in the future (39).”
Line 159; what do you mean by altered? The majority of the measurements you analyzed did not change. I did not see a single sensor, primer or activator differ between cardiac muscle in controls and DMD animals. Essentially, there was no quantifiable evidence of inflammation in cardiac muscle and some minor to moderate changes in skeletal muscle. None of which could be linked to age or disease progression.
We understand the Reviewer’s point. Thus, we have modified the Results and the Discussion sections extensively to support the main findings of our paper.
Lines 170-172; add in references.
Line 180; be consistent with using italics for the mdx.
These have been changed in the revised version of the MS.
Conclusions; your data only shows a “consistent” increase in content of IL-1beta and IL-18 in gastrocnemius muscle of mdx mice and rats. It does not provide any actual evidence of inflammasome priming and activation. Moreover, you did not run any statistics over time (age effects) and therefore you can only say there was no difference from age-matched control. Your data also does not highlight any “differences in the pathomechanisms of skeletal damage and the progression of cardiomyopathy in DMD” because you did not measure any marker of damage and regardless of changes in inflammatory markers, both tissues were fibrotic.
We appreciate the Reviewer’s insightful comment. This has been changed in the revised version of the MS accordingly. Overall, we have toned the Discussion and Conclusion sections down as well as extended the Discussion sections to improve the interpretation of our data sets.
Round 2
Reviewer 1 Report
I appreciate the authors' responses.
I have an additional comment: the quality of skeletal muscle sections presented in Figure 5 should be improved.
Author Response
Reviewer #1
I appreciate the authors' responses.
I have an additional comment: the quality of skeletal muscle sections presented in Figure 5 should be improved.
We are thankful for the positive feedback. We agree with the Reviewer's point, thus, we have included new representative images to improve the quality of the Fig.5.
Reviewer 3 Report
The authors did their best to address my concerns.
Please note that for figure 5B fibrotic area there are only 3 data points for the control, but should be 4.
Author Response
Reviewer #3
The authors did their best to address my concerns.
Please note that for figure 5B fibrotic area there are only 3 data points for the control, but should be 4.
We are grateful for the Reviewer's comments and positive feedback. We have revised and corrected the case numbers, and included new representative images to improve the quality of the Fig.5.